# Effect of Ohmic Heating on Sensory, Physicochemical, and Microbiological Properties of “Aguamiel” of *Agave salmiana*

**DOI:** 10.3390/foods9121834

**Published:** 2020-12-10

**Authors:** Luis Rascón, Mario Cruz, Rosa M. Rodríguez-Jasso, Alberto A. Neira-Vielma, Sonia N. Ramírez-Barrón, Ruth Belmares

**Affiliations:** 1Food Research Department, School of Chemical Sciences, Autonomus University of Coahuila, Saltillo Coahuila, Boulevard Venustiano Carranza e Ing. José Cárdenas, Col. República, Saltillo C.P. 25280, Coahuila, Mexico; luisfreing@hotmail.com (L.R.); rrodriguezjasso@uadec.edu.mx (R.M.R.-J.); 2Department of Food Science and Technology, Antonio Narro Autonomous Agricultural University, Calzada Antonio Narro no. 1923, Col. Buena Vista, Saltillo C.P. 25280, Coahuila, Mexico; sonia.rmz.barron@gmail.com; 3Studies and Research Interdisciplinary Center, Autonomus University of Coahuila, Saltillo Coahuila, Prolongación David Berlanga y 16 de Septiembre, Unidad Camporredondo, Saltillo C.P. 25280, Coahuila, Mexico; aneiravielma@uadec.edu.mx

**Keywords:** *Agave salmiana*, aguamiel, sensory properties, physicochemical properties, ohmic heating

## Abstract

The use of ohmic heating (OH) processing technologies in beverages might provide a higher quality value to the final product; consumers tended to prefer more natural products with minimum preservative substances. The aim of this work was to evaluate the effect of OH over the presence of microorganisms in “aguamiel” as well as to study the effects on physicochemical analysis like total sugars, soluble solids, electric conductivity pH, and color. The results showed that the conductivity of “aguamiel” was 0.374 s/m, this as temperature increased, conductivity rose as well. During OH a bubbling was observed when reaching 70 °C due to the generation of electrochemical reactions during the OH process. OH had a significant effect in the reduction of *E. coli*, yeast, and lactobacillus compared to conventional pasteurization, reaching optimal conditions for its total inactivation. Regarding its physicochemical properties, both treatments, conventional pasteurization and OH, did not show negative changes in aguamiel, demonstrating that OH technology can be a feasible option as a pasteurization technique. In conclusion it is important to notice that negative changes were not found in quality, color and sugars of “aguamiel”. Therefore, ohmic heating can be an option to replace traditional methods used for pasteurization.

## 1. Introduction

The agave commonly known as “maguey” is native to Mexico which hosts 75% of the total species in the Americas, of which 55% are endemic. Some agave species are commonly known as “maguey pulquero” (*Agave salmiana*, *A. Mapisaga*, *A. Atrovirens*, *A. Americana*, *A. Ferox*), as they contain a sap called “aguamiel” which is mainly used to produce “pulque”, one of the oldest alcoholic beverages in the Americas. Several studies have reported that agave aguamiel is a product with functional properties due to the presence of bioactive components such as amino acids and sugars [1]. Aguamiel from *Agave salmiana* is principally used to produce pulque and as as energy drink in towns where it is grown.

Aguamiel problem is the short shelf life due to its high sugar concentration and moisture, making it prone to microbial growth adding the fact that cross contamination happens during harvesting, either by contact with the environment or due to bad practices while being harvested. The beverage industry, for several years, has been using conventional heating as the only technique to pasteurization and sterilization for several years due to its low maintenance costs and because it is simple to be applied. Nevertheless, the emphasized reduction in bioactive compounds caused by the high temperatures, indicates an important loss in the beverage’s sensory and nutritional properties [2].

Because of that we are experiencing an increase in the use of emerging technologies based on electricity to be used in beverages [3], these technologies have advantages over conventional heating produced longer shelf lives in pomegranate juice [4], orange juice, apple, and melon [5]. OH has revolutionized the beverage industry as a result of its high pasteurization, sterilization, blanching and dehydration capacity [6]. OH is based on passing an electric current through a liquid or substance between electrodes, that liquid will oppose to the electric flow and its internal temperature will rise quicker than in a conventional thermic treatment, making it one of the most important techniques over the last years.

The objective of this work was to evaluate the effect of ohmic heating, in sensory, physicochemical, and microbiological properties.

## 2. Materials and Methods

### 2.1. Aguamiel Beverage

Aguamiel beverage was harvested in Guadalupe Victoria located in Saltillo, Coahuila, México. It was extracted of *A. salmiana*, filtered, placed in sterile bottles (250 mL) and stored (−18 °C) until analysis performing.

### 2.2. Ohmic Heating (OH) Equipment

The equipment used contains an autotransformer with a variable voltage, model 25/10A-14O/GVD, 10 amps. It also has a pair of stainless-steel electrodes separated by a 5 cm distance between them and a 500 mL borosilicate glass cell, as shown in Figure 1.

### 2.3. Aguamiel Electric Conductivity

The conductivity of the samples of aguamiel was measured with a HANNA instruments DiST 4 conductivity tester, 50 mL of sample were placed in ohmic cell. Three different voltages were tested (100, 150 and 200 V) with different temperature intervals (70, 80, and 90 °C), all the experiment were carried out by triplicate.

### 2.4. Ohmic and Conventional Heating

Two conventional pasteurizations were carried out, the first one being a low temperature for a long period of time (70 °C, 30 min), the second one a high temperature for a short period of time (95 °C, 5 s). Regarding the ohmic heating pasteurization, 9 treatments were carried out to optimize the inactivation of microorganisms, the details of each heat treatment are presented below.

#### 2.4.1. Conventional Heating (CH) Pasteurization Process

The CH treatment was carried out using a water bath. 25 mL of aguamiel were placed in sterile jars situated in a pot with water at room temperature. The sample reached 2 pasteurization temperatures, 95 °C for 5 s, and 70 °C for 30 min. One of the jars worked as control since it contained a thermometer to monitor the aguamiel temperature. After pasteurization the two conditions, the temperature of the jars samples were lowered to 25 °C and then refrigerated for further analysis.

#### 2.4.2. Ohmic Heating (OH) Pasteurization Process

The pasteurization process using OH was carried out in batch mode, using 178 mL of the sample (volumen to covers the electrodes). The experiment was analyzed using a three-factor analysis, using three different temperatures (70, 80, and 90 °C), voltages (100, 150, and 200 V) and time period (5, 10, and 15 s) showed in Table 1. The aguamiel was poured in the cell, which was closed using with a lid, and a thermometer was also placed inside the cell. After pasteurization, a thermal shock was carried out to lower the temperature of the aguamiel until it hit 25 °C.

### 2.5. Microbiological Analysis

The microbiological analysis was conducted according to the Official Mexican Standard (NOM), for fungi and yeast counts the NOM-111-SSA1-1994 [7], for the presence of *Escherichia coli* the NOM-210-SSA1-2014 [8], and for the sanitary specifications for various types of drinks the NOM-218-SSA1-2011 [9]. The count of *Lactobacillus* was conducted with MRS agar, petri plates were incubated at 37 °C for 5 days in anaerobiosis with the count made afterwards for both crude and treated samples.

### 2.6. Physicochemical Analysis: pH, Total Sugars and Soluble Solids

In order to conduct the pH measurement, a potentiometer (OAKLON by EUTECH instruments) was used. Soluble solids measurement were made with an automatic refractometer (hand-held ATAGO) was used, calibrated using distilled water; results were shown as Brix degrees. Lastly, the analysis of sugars was conducted using an anthrone test. Crude and treated samples were analyzed in triplicate.

### 2.7. Color Analysis

A Precision Colorimeter NR20XE was used in both the crude and treated samples. For this, 25 mL of sample were placed in a transparent container on a white tray; 5 repetitions for each sample were measured in several areas to be reproducible. The colorimeter values were lightness (*L**), redness (*a**), yellowness (*b**) chromaticity and hue.

### 2.8. Statistical Analysis

An analysis of variance (ANOVA) was performed along with Tukey’s tests to establish statistically significant differences between means of all variables, samples were analyzed by three independent batches of aguamiel and from each batch three samples were analyzed. Analyses were performed using Minitab 17 and Infostat 2016 software (Infostat, Córdoba, Argentina).

## 3. Results and Discussion

### 3.1. Evaluation of Electric Conductivity and Temperature of Aguamiel Samples

The temperature curves of aguamiel during ohmic heating at different voltages are showed in Figure 2. The results show that aguamiel is a good electrical conductor due to its high content of ions which makes it vulnerable to electric current.

It can be seen in Figure 3, that at higher voltage (200 V) temperature increases faster with a time of 70 s, followed by the voltages 150 and 100 with times of 96 and 250 s respectively, therefore, the heating times were reduced due to the high electrical force applied. Also was that at higher voltage (30–40 V/cm^−1^) temperature rose faster in orange and tomato juices, in both studies the initial temperature was room temperature [10].

In addition, bubble formation was observed when reaching temperatures above 60 °C, when the voltage gradient increased, this same phenomenon was reported with tomato juice [11]; bubbling could have been originated by electrochemical reactions that release gas [12]. In addition, during bubbling, electrical conductivity no longer increased as it did initially as solids become more concentrated due to evaporation of aguamiel [13]. Changes of electrical conductivity due to temperature can be found in Figure 3. While temperature was increased, the electrical conductivity, showed a maximum point of 0.616 s/m with a voltage of 150 v, meaning that the higher the voltage, the higher the electrical conductivity in the aguamiel. It should be noted that aguamiel has an electrical conductivity of 0.374 s/m with an acidic pH and temperature of 24 °C, compared to beer and coffee its conductivity is higher [14]. Similar results were observed regarding the speed of electrical conductivity, a possible explanation would be that this is due to the increase of ionic or molecular agitation of aguamiel, therefore increasing electrical conductivity more [15,16]. Some studies reported minerals presents in aguamiel such as calcium (Ca), phosphorus (P), magnesium (Mg), zinc (Zn), iron (Fe), copper (Cu) and boron (Bo), favoring these obtained results [17] these ions help an increase in electrical conductivity.

### 3.2. Microbiological Analysis

The content of microorganisms in aguamiel as well as comparison of their reduction by conventional and ohmic heating treatments are found in Table 2. The presence of *E. coli* 3.0 × 10^5^ CFU/mL, yeast 2.86 × 10^5^ CFU/mL and *Lactobacillus* 5.4 × 10^4^ CFU/mL was found in raw aguamiel (RA). Probably the presence of *E. coli* was result of contamination during the harvest of aguamiel, possibly because of the environment, contact of the harvester with the sample, and waste from plants or animals. Several works have been reported regarding the relevance of beneficial microorganisms taken from samples of agave.

Table 2 shows the reduction of *E. coli* by comparing the two conventional treatments versus ohmic treatments. The results show that both conventional pasteurization treatments had a significant reduction of *E. coli* as expected, but if it presented a high amount of yeast and *Lactobacillus* with 70 °C 30 min, these microorganisms were responsible for characteristics such as odor and flavor in foods [18]. The 9 treatments using ohmic heating also presented a significant effect against *E. coli*, except treatment POH-1 in which 200 CFU/mL of *E. coli* survived nevertheless to aguamiel is pathogen free, this result could be due, there are several factors that might contaminated aguamiel samples for example the handling, analysis or transport conditions.

On the other hand, the effect of OH treatments presented an inactivation of pathogenic microorganisms, yeasts, and *Lactobacillus;* therefore, there are few studies regarding the reduction of *Lactobacillus* as they possess probiotic properties; in the case of aguamiel, it contains plenty of these bacteria which accelerate fermentation, and in order to ensure the quality of aguamiel, it must be inactivated. There were changes reported in microbial morphology due to the use of OH in both yeast and *Lactobacillus* as a result to a phenomenon called electroporation [19].

Our result presented a microbiological inactivation under the study conditions, but other investigations evaluated the effect of ohmic heating at low temperatures (50–60 °C) in apple juice and peptone water to reduce *E. coli* O157:H7, *Salmonella,* and *Listeria monocytogenes*, and found a significant reduction in all three pathogens compared to conventional treatment [20]. Additionally, ohmic heating also reduced to zero the presence of *Salmonella*, coliforms and *Staphylococcus* in the juice of *Aloe vera* [12]. In orange juice, a faster reduction of *E. coli* O157:H7 was found at a more acidic pH in combination with the effect of ohmic heating [10]; another study evaluated the effect of ohmic heating in tomato and orange juice and the results were positive regarding the reduction of pathogenic bacteria such as *E. coli* O157:H7, *S. typhimurium* and *L. monocytogenes* [21]. In addition, similar results were also obtained when was applying an electrical force over *Aloe vera* juice, finding a significant reduction on mesophilic aerobic microorganisms as well as guaranteeing the safeness of the drink [22].

Therefore, OH had a lethal effect on *Lactobacillus*, reducing its population to zero in all treatments using it, while in the one using pasteurization for 30 min at 70 °C there was a survival of 960 CFU/mL, the survival of *Lactobacillus* colonies in aguamiel does not have a lethal effect on health causing an effect over the quality of the drink, recalling it offers a probiotic function but highlighting aguamiel contains fructooligosaccharides (FOS) providing its functionality.

Likewise, a reduction to zero in molds and yeasts was found with the OH treatments, while the conventional treatment at 70 °C for 30 min was the one with the lowest lethality, surviving 2100 CFU/mL, therefore, OH has a lethal effect on the microorganisms present in aguamiel, presenting the phenomenon of electroporation when compared to conventional pasteurization, causing an increase in the permeability of the bacterial cell membrane due the increase of pore size and their immediate death [23]; in addition to its rapid increase in temperature, making it an effective method [14].

In other investigations there were found a greater reduction of molds and yeasts in buffalo milk due to the effect of OH, while in a conventional heating treatment there was a greater survival of these microorganisms [24]. In contrast, it did not detect the presence of molds and yeasts in the sugarcane juice due to the effect of OH [25], the same effect in papaya pulp [26], in orange juice [27], and in *Aloe vera* juice [22].

### 3.3. Analysis of Physicochemical Properties: pH, Total Sugars and Soluble Solids

The results of the physicochemical analyses are shown in Table 3, indicating the comparison between raw aguamiel, aguamiel treated with conventional pasteurization and ohmic heating. Differences were not significant in pH measurement, highlighting those treatments that resemble raw aguamiel with a pH of 6.2, which are POH-4, POH-5, POH-7, POH-8, and POH-9, meaning that this important parameter of aguamiel was maintained. Meanwhile the other treatments such as the pasteurization at 70 °C 30 min, 95 °C 5 s and POH-1 had a slight decrease in pH. In addition, there was an increase in pH level in POH-2 and POH-3. The slight changes in pH, whether high or low, happen due to the heating rate of OH, while a higher heating rate is less the change in pH, meaning hydrolysis reactions are carried out in short time [13,28]. Acidic pH less than 6, indicates the presence of bacteria indicating spontaneous aguamiel fermentation. The pH value was almost constant which significantly affected the quality of aguamiel, yet there was no degradation.

The results of the analysis of soluble solids are shown in °Brix and significant differences can be observed between raw aguamiel and samples after OH treatment, only the 70 °C 30 min treatment did not show any difference. There was an increase in soluble solids after OH, a possible explanation is the intense bubbling that was observed during OH, evaporating the aguamiel and increasing the concentration of some solids such as sugars or proteins [29]. Significant differences were found in all pasteurization treatments in comparison with raw aguamiel, standing out the treatments that were similar to the control, such as the 95 °C 5 s treatment, POH-4, POH-5, POH-7, POH-8, and POH-9. This trend indicates that voltage was not what caused the degradation of sugars, but temperature was the one which did. On the other hand, the treatment at 70 °C 30 min was the one that had the greatest degradation of sugars, a possible explanation is that a process of caramelization occurred during pasteurization. OH treatments did not affect these compounds per se, similar results were found in the increase of total sugars in the plum jam [29].

### 3.4. Color Analysis

Color is an important parameter in food; hence it is important to consider this after pasteurization treatment, in aguamiel if a color gives a white hue it means that it has fermented. The results of the luminosity, hue and chromaticity of raw aguamiel versus the samples treated with conventional pasteurization and OH are found later.

The results showed below present many differences among the studied values and do not indicate a trend toward a variable that affects color quality itself. The luminosity (*L**) is presented in the Figure 4, it was found that 5 of the 9 treatments made with OH (POH-1, POH-2, POH-3, and POH-9) are statistically equal to raw aguamiel, keeping *L**. This same effect was reported in orange juice [21], in tomato juice [30] which maintained the color after OH.

In the other treatments (70 °C 30 min, POH-4, POH-5, POH-6, POH-7, POH-8) *L** value decreased, the treatment of 95 °C 5 s had an increased in luminosity. We found that as the frequency of OH increases, so does color degradation in acerola cherry pulp, and that it has a correlation with bioactive compounds; in this case there was no great color degradation in aguamiel, so it can be said that nutrients were kept [31]. Also, we found a slight decrease in *L** by OH [10], but with correlation with an acid pH and an overall color decrease in orange and pineapple juices [32].

Regarding tone, it was found that most of the treatments maintained the yellow tone, even on higher temperature the sample kept a yellow tone; as shown in Figure 5, the 100 V voltage sample was the one with the highest retention of the yellow hue compared to the crude sample.

Changes in chromaticity are shown in Figure 6 where it can be seen that the 95 °C 5 s treatment had a more intense color than the rest, even higher than in the crude sample. Yet, it also indicates that the higher the temperature, chromaticity improves. Statistically, raw aguamiel was different from the other treatments since, for the most part, samples treated with OH (POH-1, POH-2, POH-3, POH-7, POH-8, and POH-9) had a more intense color whereas in the other treatments (70 °C, 30 min, POH-4, POH-5, and POH-6) the samples had muted colors. Similar results were found in orange juice, with high chromaticity after the OH process [33].

## 4. Conclusions

An increase in electrical conductivity may be due to the fact that aguamiel contains numerous ions that help to increase the rate of heating as the voltage increases, decreasing to more than 3 times the time necessary to reach the pasteurization temperature of 95 °C, using 200 V, therefore, ohmic heating represents an innovative alternative in the production processes of functional beverages such as aguamiel. Microbiologically, the electrical conductivity is lethal for the microorganisms present in aguamiel; this can guarantee the safety of beverages with high concentrations of sugars that present high microbial loads. *Lactobacillus* were the most susceptible to the applied electrical force, it was also found that 80 °C, 200 V for 5 s are the optimal parameters necessary for the inactivation of microorganisms in aguamiel.

The color characteristics are not drastically affected using ohmic heating due to the short time that the process takes, avoiding the darkening due to the caramelization of the sugars present in aguamiel, and maintaining the characteristic color of the drink.

Due to the above, ohmic heating is an option to replace the traditional methods used to pasteurize beverages such as aguamiel.

## Figures and Tables

**Figure 1 foods-09-01834-f001:**
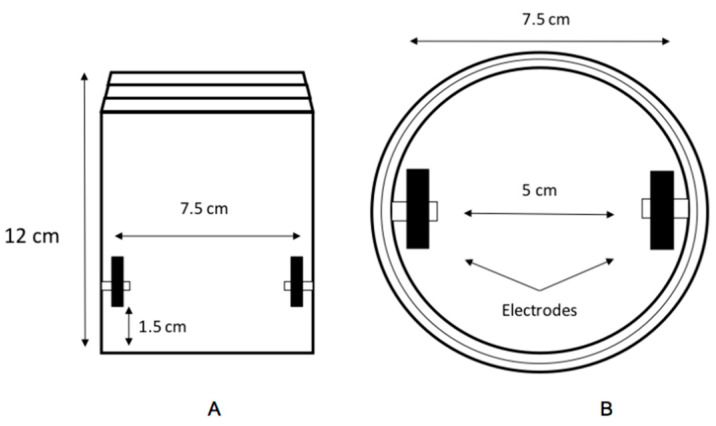
Diagram of the ohmic cell characteristics: (**A**) Ohmic cell general overview; (**B**) Top view of the ohmic cell.

**Figure 2 foods-09-01834-f002:**
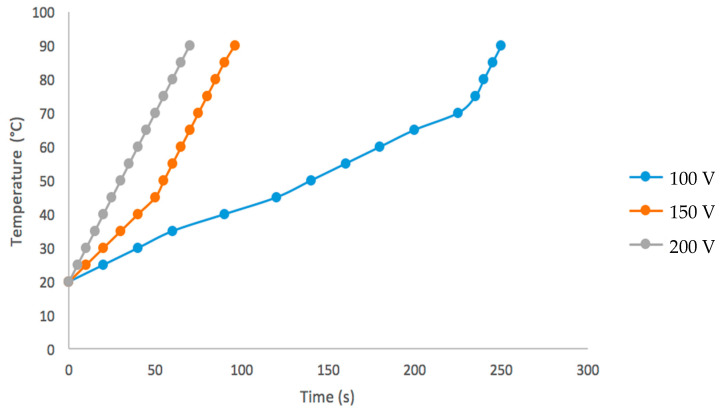
Evaluation of the temperature of aguamiel during ohmic heating at different voltages.

**Figure 3 foods-09-01834-f003:**
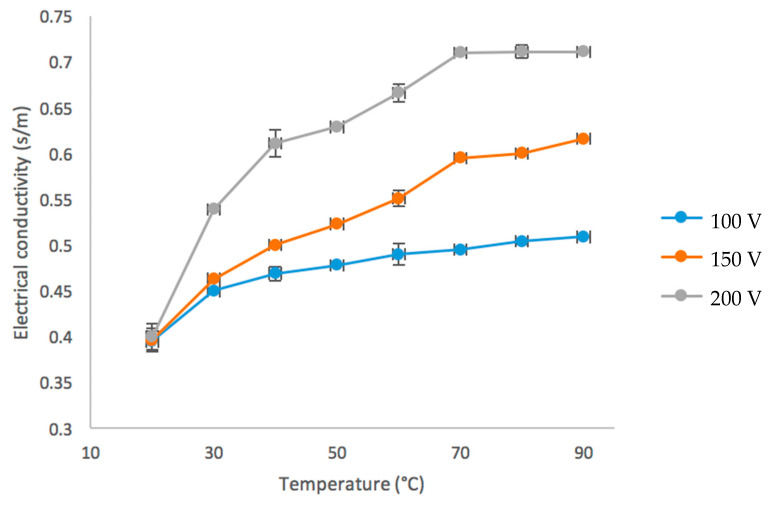
Changes in electrical conductivity given to temperature of aguamiel at different voltages.

**Figure 4 foods-09-01834-f004:**
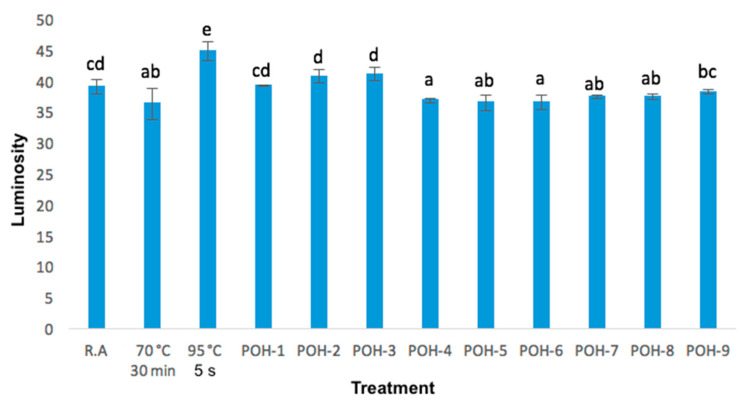
Luminosity changes in raw aguamiel, pasteurized aguamiel (70 °C, 30 min), pasteurized aguamiel (95 °C, 5 s) and ohmic heating treatments; data is mean (±standard deviation) of three replicates. a, b, c, d, e different letters indicate significant differences (*p* < 0.05).

**Figure 5 foods-09-01834-f005:**
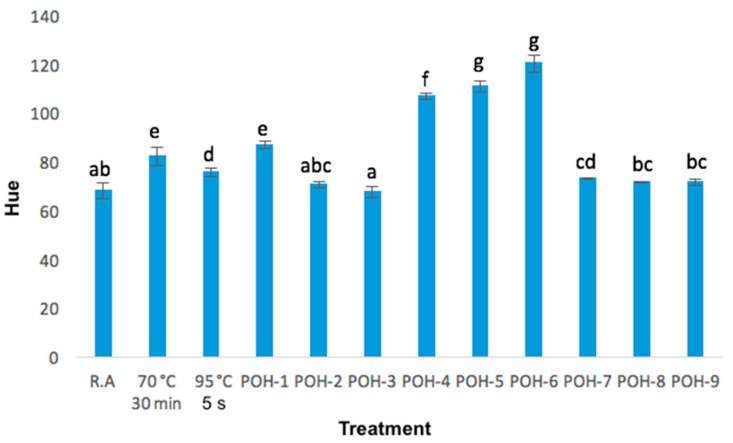
Changes in the hue in raw aguamiel and after pasteurization and ohmic heating treatments; data is mean (±standard deviation) of three replicates. a, b, c, d, e, f, g letters indicate significant differences (*p* < 0.05).

**Figure 6 foods-09-01834-f006:**
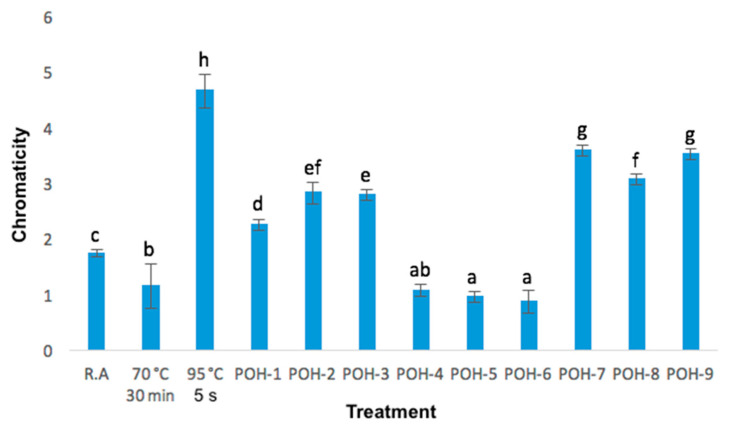
Changes in chromaticity in raw aguamiel and after pasteurization and ohmic heating treatments; data is mean (±standard deviation) of three replicates. a, b, c, d, e, f, g, h letters indicate significant differences (*p* < 0.05).

**Table 1 foods-09-01834-t001:** Conditions for Ohmic Heating optimization.

Treatments	Temperature (°C)	Voltage (V)	Time (s)
POH-1	70	100	5
POH-2	70	150	10
POH-3	70	200	15
POH-4	80	100	5
POH-5	80	150	10
POH-6	80	200	15
POH-7	90	100	5
POH-8	90	150	10
POH-9	90	200	15

**Table 2 foods-09-01834-t002:** Microbiological analysis of raw and pasteurized aguamiel by conventional and ohmic heating.

Treatment	CFU/mL		
	*E. coli*	Yeasts	Lactobacillus
Raw aguamiel	3.0 × 10^5^	2.86 × 10^5^	5.4 × 10^4^
95 °C 5 s	n.a*	-	-
70 °C 30 min	n.a*	2100	960
POH-1	200	-	-
POH-2	n.a*	-	-
POH-3	n.a*	-	-
POH-4	n.a*	-	-
POH-5	n.a*	-	-
POH-6	n.a*	-	-
POH-7	n.a*	-	-
POH-8	n.a*	-	-
POH-9	n.a*	-	-

The (n.a*) means absent. The (-) means no growth.

**Table 3 foods-09-01834-t003:** Physicochemical analysis of raw aguamiel, conventionally pasteurized aguamiel and ohmic heating aguamiel.

	pH	Soluble Solids (°Brix)	Total Sugars
Raw aguamiel	6.2 ± 0.1 ^b^	9.3 ± 0.5 ^a^	147.67 ± 1.1 ^abc^
70 °C 30 min	6.2 ± 0.2 ^a^	8.8 ± 0.1 ^a^	95.59 ± 1.9 ^h^
95 °C 5 s	6.1 ± 0.1 ^a^	10 ± 0.1 ^b^	164.60 ± 7.8 ^a^
POH-1	6.1 ± 0.1 ^a^	11 ± 0.2 ^c^	105.94 ± 1.1 ^fg^
POH-2	6.5 ± 0.2 ^c^	11 ± 0.1 ^c^	104.24 ± 2.8 ^fg^
POH-3	6.5 ± 0.2 ^c^	10 ± 0.2 ^c^	121.50 ± 1.3 ^ef^
POH-4	6.2 ± 0.1 ^b^	11 ± 0.1 ^c^	158.02 ± 3.3 ^ab^
POH-5	6.2 ± 0.2 ^b^	11 ± 0.5 ^c^	125.63 ± 2.7 ^cdef^
POH-6	6.1 ± 0.2 ^a^	11 ± 0.5 ^c^	122.50 ± 2.7 ^def^
POH-7	6.2 ± 0.2 ^b^	11 ± 0.5 ^c^	150.49 ± 4.2 ^ab^
POH-8	6.2 ± 0.1 ^b^	11 ± 0.6 ^c^	144.54 ± 1.5 ^abcd^
POH-9	6.2 ± 0.2 ^b^	12 ± 0.5 ^d^	137.85 ± 1.8 ^bcde^

Data is mean of three replicates (± standard deviation). Different letters (a, b, c, d, e, f, g, h) indicate significant differences within the same column (*p* < 0.05).

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
