# Peer review of "Effect of Ohmic Heating on Sensory, Physicochemical, and Microbiological Properties of “Aguamiel” of Agave salmiana"

_foods, 2020, doi:10.3390/foods9121834_

Round 1
Reviewer 1 Report
The authors of the manuscript "Effect of ohmic heating on sensory, physicochemical and microbiological properties of ¨aguamiel¨ of Agave salmiana" investigated the possibility of using ohmic heating to stabilize "aguamiel". The manuscript undoubtedly has practical value. However, I have a few questions for the authors:
1) Is it possible to heat 25 ml of sample to 95 degrees Celsius in just 5 seconds? This of course depends on the specific heat capacity, but it seems that 5 seconds is not enough to heat the sample using a water bath.
2) Please explain the selection of the experiment factors. The scheme is not full (3x3x3). If it were, a three-factor analysis would be possible.
3) Why are the significance of the differences not included in the graphs in section 3.4?
Author Response
November, 2020
PROF. DR. THEODOROS VARZAKAS,
We have revised and attended the comments made by reviewers on the manuscript titled "Effect of ohmic
heating on sensory, physicochemical and microbiological properties of ¨aguamiel¨ of Agave salmiana"
(foods-1000771).
On behalf of our research group, I would like to thank reviewers and editors for their valuable comments
and time. We are confident the manuscript has been improved by your insight. Changes done on the
manuscript have been highlighted and can be easily undone if required in further processing.
In the table below we address each comment made and our response:
Reviewer #1
1. Is it possible to heat 25 ml of sample to 95 degrees Celsius in
just 5 seconds? This of course depends on the specific heat
capacity, but it seems that 5 seconds is not enough to heat the
sample using a water bath.
It was corrected because the sample was
the one that reached that temperature. Line
83-88
2. Please explain the selection of the experiment factors. The
scheme is not full (3x3x3). If it were, a three-factor analysis
would be possible.
The change has been done. Line 91-92
3. Why are the significance of the differences not included in the
graphs in section 3.4?
The change has been done. Line 249
Sincerely,
RUTH BELMARES

Reviewer 2 Report
The paper entitled "Effect of ohmic heating on sensory, physicochemical and microbiological properties of ¨aguamiel¨ of Agave salmiana " describes the use of ohmic heating for increasing the safety of "aguamiel". The purpose of the paper is very interesting nevertheless the content should be more clearly discuss and focused on the results obtained and not so much in the literature review. Also the English language used does not make easy to read the paper
Detailed comments
they produce longer shelf lives,
where 50 ml of sample were placed on it and the tester was dunked.
Minerals presents in 142 aguamiel are calcium (Ca), phosphorus (P), magnesium (Mg), zinc (Zn), iron (Fe), copper (Cu) and 143 boron (Bo), favoring these obtained results [17].
Table 2. Microbiological analysis of crude aguamiel and pasteurized aguamiel by conventional and 155 ohmic heating, (n.a) not applicable. Why not applicable to E. coli?
The results show that both conventional pasteurization treatments had a 158 significant effect against E. coli and as expected, most of the 9 treatments using ohmic heating also 159 presented a significant reduction in E. coli.????
The Official Mexican Standard [9] for non-alcoholic beverages indicates that the allowed number 161 of UFC/ml for E. coli is not applicable (n.a), thus the treatments used in this study meet the 162 specifications of the Standard, except treatment POH-1 in which 200 CFU/ml of E. coli survived.
What do you mean by not applicable? It is not present or not analysed? But you found E. coli in the initial sample. N.a. means that no colony was detected?
Significant differences were found in the pH measurement: value?
Regarding tone: Tone or hue???
The results show that aguamiel contains numerous ions which help an increase in electrical 261 conductivity, thus reducing processing time: you only show that it has electrical conductivity as you did not performed the analysis of the mineral content
Therefore I think that the paper should be revised carefully
Author Response
RUTH BELMARES CERDA
UNIVERSITY OF COAHUILA
Boulevard Venustiano Carranza sin número,
Colonia República, Saltillo, C.P. 25280.
Coahuila Mexico
November, 2020
PROF. DR. THEODOROS VARZAKAS,
We have revised and attended the comments made by reviewers on the manuscript titled "Effect of ohmic
heating on sensory, physicochemical and microbiological properties of ¨aguamiel¨ of Agave salmiana"
(foods-1000771).
On behalf of our research group, I would like to thank reviewers and editors for their valuable comments
and time. We are confident the manuscript has been improved by your insight. Changes done on the
manuscript have been highlighted and can be easily undone if required in further processing.
In the table below we address each comment made and our response:
Reviewer #2
1. The paper entitled "Effect of ohmic heating on sensory,
physicochemical and microbiological properties of
¨aguamiel¨ of Agave salmiana " describes the use of
ohmic heating for increasing the safety of "aguamiel". The
purpose of the paper is very interesting nevertheless the
content should be more clearly discuss and focused on the
results obtained and not so much in the literature review.
Also the English language used does not make easy to
read the paper.
The changes has been done, sentences rephrased
and attended the reviews.
English language has been corrected
2. Detailed comments:
They produce longer shelf lives,
Where 50 ml of sample were placed on it and the tester
was dunked.
Minerals presents in aguamiel are calcium (Ca),
phosphorus (P), magnesium (Mg), zinc (Zn), iron (Fe),
copper (Cu) and boron (Bo), favoring these obtained
results [17].
The changes has been done, sentences rephrased
and attended the reviews.
3. Table 2. Microbiological analysis of crude aguamiel and
pasteurized aguamiel by conventional and ohmic heating,
(n.a) not applicable. Why not applicable to E. coli?
The Official Mexican Standard [NOM-218-
SSA1-2011] for non-alcoholic beverages
indicates that the allowed number of UFC/ml for
E. coli is not applicable (n.a), means absent. L
156
4. The results show that both conventional pasteurization
treatments had a significant effect against E. coli and as
expected, most of the 9 treatments using ohmic heating
also presented a significant reduction in E. coli.????
The change has been done, L158-165
5. The Official Mexican Standard [9] for non-alcoholic
beverages indicates that the allowed number of UFC/ml
for E. coli is not applicable (n.a), thus the treatments used
in this study meet the specifications of the Standard,
except treatment POH-1 in which 200 CFU/ml of E. coli
survived.
Sentence rephrased, L158-165
6. What do you mean by not applicable? It is not present or
not analysed? But you found E. coli in the initial sample.
N.a. means that no colony was detected?
The UFC/ml for E. coli is not applicable (n.a),
means absent according to the NOM-218-SSA1-
2011.
7. Significant differences were found in the pH
measurement: value?
Sentence rephrased: Differences were not
significant in pH measurement L207
8. Regarding tone: Tone or hue??? Done, is Hue
9. The results show that aguamiel contains numerous ions
which help an increase in electrical conductivity, thus
reducing processing time: you only show that it has
electrical conductivity as you did not performed the
analysis of the mineral content
The sentence has been reformulated. L268
Effectively not performed the analysis of the
mineral content
Sincerely,
RUTH BELMARES

Reviewer 3 Report
Interesting study with practical application, on the use of innovative processing technologies to preserve a tradition beverage.
The aim of the study is clearly defined.
Regarding the experimental design, how many independent "aguamiel" beverage batches did the authors analyse? And how many "aguamiel" samples per batch?
Do the authors consider the presence of Lactobacillus in the final product as harmful? Wouldn't their presence have beneficial effects due to their probiotic properties? Or do the authors want to stop the fermentation and stabilise the final product?
References are adequate and up-to-date.
Author Response
RUTH BELMARES CERDA
UNIVERSITY OF COAHUILA
Boulevard Venustiano Carranza sin número,
Colonia República, Saltillo, C.P. 25280.
Coahuila Mexico
November, 2020
PROF. DR. THEODOROS VARZAKAS,
We have revised and attended the comments made by reviewers on the manuscript titled "Effect of ohmic
heating on sensory, physicochemical and microbiological properties of ¨aguamiel¨ of Agave salmiana"
(foods-1000771).
On behalf of our research group, I would like to thank reviewers and editors for their valuable comments
and time. We are confident the manuscript has been improved by your insight. Changes done on the
manuscript have been highlighted and can be easily undone if required in further processing.
In the table below we address each comment made and our response:
Reviewer #2
1. Regarding the experimental design, how
many independent "aguamiel" beverage
batches did the authors analyse? And how
many "aguamiel" samples per batch?
The changes has been done.
L115
2. Do the authors consider the presence
of Lactobacillus in the final product as
harmful?
No. However the OH decreases the microbial load of the
beverage and does not discriminate whether they are
beneficial as Lactobacillus or pathogens such as E. coli.
L: 185-189
3. Wouldn't their presence have beneficial
effects due to their probiotic properties?
Yes. The lactobacillus are probiotics
L: 185-189
4. Or do the authors want to stop the
fermentation and stabilise the final product?
The functionality also depends on the chemical composition
of aguamiel for example FOS
L: 191-197
Sincerely,
RUTH BELMARES

Round 2
Reviewer 1 Report
The authors significantly improved the manuscript. However, not all issues have been clarified in my opinion, especially regarding the design of the experiment. However, the manuscript contains valuable data.
Author Response
Data is mean (±standard deviation) of three replicates. different letters indicate significant differences (p < 0.05) theses changes be seen in figures of section 3.4

Reviewer 2 Report
Althoug the author made substantial revision of the paper according to the reviewers suggestions, there are still some points that should be addressed in order to improve the manuscript.
279 Regarding tone, replace tone by hue and also in all the manuscript
287 had a more intense color than the rest, even better than in the crude sample. Better or higher?
Please rewritte conclusions
The initial sentence was not a conclusion it was an observation that explains the low processing times. This sentence is generic and is not the obtained from the results in your work.
An increase in electrical conductivity may be due because aguamiel 299 contains numerous ions which help an increase in electrical conductivity, thus reducing processing 300 times, furthermore the effect of ohmic heating is lethal on the microorganisms present in aguamiel;
If you have a 3X3 factorial design why don’t you use the results obtained by statistical analysis (ANOVA) to discuss the effects of each factor: Temperature/Voltage/Time and you do a discussion comparing point by point. This would greatly improve the clarity of the manuscript and the effect of each variable in the final result
Author Response
|
1. Regarding tone, replace tone by hue and also in all the manuscript |
Done. |
|
2. Had a more intense color than the rest, even better than in the crude sample. Better or higher? |
Done. |
|
3. Please rewritte conclusions |
Done. |
|
4. The initial sentence was not a conclusion it was an observation that explains the low processing times. This sentence is generic and is not the obtained from the results in your work. |
Sentence rephrased |
|
5. An increase in electrical conductivity may be due because aguamiel 299 contains numerous ions which help an increase in electrical conductivity, thus reducing processing 300 times, furthermore the effect of ohmic heating is lethal on the microorganisms present in aguamiel; |
Sentence rephrased in conclusions |
|
6. If you have a 3X3 factorial design why don’t you use the results obtained by statistical analysis (ANOVA) to discuss the effects of each factor: Temperature/Voltage/Time and you do a discussion comparing point by point. This would greatly improve the clarity of the manuscript and the effect of each variable in the final result |
An analysis of variance (ANOVA) was performed along with Tukey’s tests to establish statistically significant differences between means of all variables (section 2.8) |

Reviewer 3 Report
In lines 120-121, the authors state that "samples were analyzed by triplicate in batches of three".
Do they mean that three independent batches of aguamiel were produced and from each batch three samples were analysed?
Or did they produce only one batch of aguamiel and three samples were analysed in triplicate (three times/three determinations)?
It is important to clarifiy this issue in the paper before drawing any conclusions.
Author Response
|
1. In lines 120-121, the authors state that "samples were analyzed by triplicate in batches of three".
Do they mean that three independent batches of aguamiel were produced and from each batch three samples were analysed?
Or did they produce only one batch of aguamiel and three samples were analysed in triplicate (three times/three determinations)?
It is important to clarifiy this issue in the paper before drawing any conclusions. |
The changes has been done. L115-116 |
